# Bluetongue Virus Infection of Goats: Re-Emerged European Serotype 8 vs. Two Atypical Serotypes

**DOI:** 10.3390/v14051034

**Published:** 2022-05-13

**Authors:** Christina Ries, Martin Beer, Bernd Hoffmann

**Affiliations:** Institute of Diagnostic Virology, Friedrich-Loeffler-Institut, Südufer 10, 17943 Greifswald-Insel Riems, Germany; christina.ries@fli.de (C.R.); martin.beer@fli.de (M.B.)

**Keywords:** atypical BTV, pathogenesis, horizontal transmission, animal experiment, goats, BTV-25, BTV-33, BTV-8

## Abstract

In recent years, numerous atypical Bluetongue virus (BTV) strains have been discovered all around the world. Atypical BTV strains are phylogenetically distinct from the classical BTV serotypes 1–24 and differ in terms of several biological features. For the first time, the atypical strains BTV-25-GER2018 and BTV-33-MNG3/2016 as well as the re-emerged classical strain BTV-8-GER2018 were evaluated comparatively in a pathogenesis study in goats—the natural host of atypical BTV. A substantial number of in-contact animals were included in this study to detect potential contact transmissions of the virus. After infection, EDTA blood, ocular, nasal and oral swab samples as well as serum were collected regularly and were used for virological and serological analyses, respectively. Our study showed differences in the immunological reaction between the two atypical BTV strains (no group-specific antibody detection) and the classical BTV strain BTV-8-GER2018 (group-specific antibody detection). Furthermore, we observed an increase in the total WBC count (neutrophils and lymphocytes) in goats infected with the atypical BTV strains. No horizontal transmission was seen for all three strains. Our study suggests that the atypical BTVs used in the trial differ from classical BTVs in their immunopathogenesis. However, no evidence of direct contact transmission was found.

## 1. Introduction

Bluetongue virus (BTV), as part of the virus family *Reoviridae* and the genus *Orbivirus,* is characterised by a double-stranded RNA genome divided into ten segments of different sizes [1]. Among the seven structural (VP1–7) and six non-structural proteins (NS1–NS5, NS3a), VP2 plays the most important role for serotype specificity by carrying the neutralising epitopes [2,3,4]. Serotype affiliation to 1 of the 24 notifiable classical BTV serotypes needs to be examined by the reference method—the virus neutralisation test [5]. However, traditional serotyping is impeded in the case of atypical BTV, and molecular typing of segment 2/VP2 leads to the classification of putative novel serotypes. Up to now, 36 (putative) serotypes—24 serotyped notifiable serotypes, 3 more or less ‘serotyped’ atypical serotypes and 9 atypical putative novel serotypes—have been described [6,7]. The arthropod-borne virus (arbovirus) is transmitted via infected female *Culicoides* and causes bluetongue disease in wild and domestic ruminants [8,9]. The clinical outcome of a BTV infection is highly variable depending not only on the BTV strain but also on the host’s breed, genetics, age and immune status [10]. Bluetongue disease, known as a major disease of sheep, particularly affects European wool sheep breeds, with high mortality and morbidity rates [11,12]. Bluetongue disease is a systemic haemorrhagic viral fever, and clinical symptoms derive from direct viral endothelial cell damage and host cell response. Oedema in the head region is common, and pulmonary oedema can be fatal for the affected animal [13,14,15]. In contrast to sheep, cattle and goats are mostly known to develop subclinical disease [16]; nevertheless, the morbidity in cattle during the first BTV-8 incursion was estimated to be around 10% [14].

In northern European countries, two important BTV incursions have been reported up to now, both caused by BTV-8 strains. With the help of animal movement restrictions and large-scale vaccination campaigns, it could be possible to successfully eradicate the disease from northern European [17,18,19]. Thus, Germany was declared officially BTV-free from February 2012 until December 2018 [20,21]. Then, the first BTV-8 case to re-emerge in Germany was detected. Over the following years, only a few additional BTV-8 outbreaks were noted in healthy cattle and calves. The BTV-8 strain that re-emerged caused mostly mild or no clinical symptoms [22]. This is in line with the findings of an experimental study in sheep, which showed that this BTV-8 strain was less pathogenic than the BTV-8 strain circulating during the first BTV-8 epidemic of 2006–2009 [23]. The accidental release of frozen material contaminated by bull semen infected with the first BTV-8 epidemic strain was suggested to have caused this re-emergence [24].

Atypical BTV, also known as small ruminant or goat-associated BTV, differs from the classical BTV serotypes in terms of both molecular and biological features [15]. None of the atypical strains have been reported in cattle [25]. For the classical BTV strains, a prolonged viraemia of <60 days has been reported, whereas for the Swiss BTV-25-TOV (Toggenburg virus) and German BTV-25-GER2018 strain, a viraemia of several years has been described [26,27]. Moreover, infection with classical BTV strains led to a long-lasting and protective humoral response, with antibodies formed against VP2 and VP7. In the virus neutralisation test (VNT), neutralisation of the classical virus strain with the respective reactive serum can be observed. BTV-25-GER2018 and the three Mongolian strains showed only partial neutralisation in the VNT in contrast to the classical strains. Several atypical BTV strains do not grow on vector-derived *Culicoides* cell lines, and animal experiments have suggested vector-independent direct contact transmission for BTV-26, -27 and -28 [28,29]. Field infections with atypical BTV strains mostly remained without clinical disease. Mild clinical disease was observed after the experimental infection of sheep with BTV-25 and BTV-26 [30], and moderate clinical signs were observed in sheep experimentally infected with BTV-28 [29].

We performed an animal experiment in goats, as they represent the natural host of atypical BTVs. With the high number of in-contact goats, we could further analyse the contact transmission as one route of horizontal transmission. Two novel atypical BTV strains (BTV-25-GER2018 and the putative novel BTV-33-MNG3/2016), both isolated from clinically healthy goats, were chosen for the animal trial. BTV-25-GER2018 originated from southern Germany [27], and the BTV-33-MNG3/2016 strain originated from Mongolia [7]. The classical BTV strain—the BTV-8-GER2018 strain that re-emerged—was isolated in Germany in 2018 from cattle [21] and was, for the first time, tested in an animal experiment with goats.

## 2. Materials and Methods

### 2.1. Animals

Thirty 4- to 6-month-old Thuringian goats (twenty-six female and four castrated male goats) were kept at the facilities of the Friedrich-Loeffler-Institut, Insel Riems, Germany, under biosafety level 3 conditions. The competent authority (State Office for Agriculture, Food Safety and Fisheries of Mecklenburg-Vorpommern, Rostock, Germany; Ref. No. LALLF 7221.3-1-048/19; date of approval 7 November 2019) approved the animal experiment. For the study, goats from a restriction-free area with no history of BTV vaccination were chosen. Before housing at the FLI, all goats tested negative for the presence of BTV antibodies in cELISA and for BTV genome by RT-qPCR. All animals were in good health. 

### 2.2. Virus Preparations

Two atypical BTV strains (BTV-25-GER2018 and BTV-33-MNG3/2016), both isolated from clinically healthy goats, as well as BTV-8-GER2018, were chosen for the animal trial. The full-length sequences of the coding regions for all three virus isolates are available (BTV-25: LR798441-50; BTV-33: LR877358-67; BTV-8: OM523087-96). The BTV-33-MNG3/2016 strain was isolated on BHK-21 (CT) cells (FLI cell culture collection number RIE0164) and then passaged five times on BHK-21 (BSR/5) cells (FLI cell culture collection number RIE0194). BHK-21 (BSR/5) cells are a clone of BHK-21 cells. The BTV-25-GER2018 strain was isolated on BHK-21 (BSR/5) cells and passaged a total of six times on BHK-21 (BSR/5) cells, whereas the BTV-8-GER2018 strain was isolated on BHK-21 (BSR/5) cells and then passaged again twice on BHK-21 (BSR/5) cells. All the virus stocks contained the infected cell–supernatant mixture of the last previously described cell passage and were stored at −80 °C until usage for the animal trial.

### 2.3. Experimental Design and Sample Collection

Three groups (A, B, C), each with ten randomly assigned animals, were kept in separate rooms without contact. In each group, five of the ten goats were inoculated subcutaneously in the shoulder neck region with 4 mL of the respective virus preparation at two different injection sites. The other five goats were kept in direct contact within the same housing unit as transmission controls. In group A, five goats were infected with BTV-33-MNG3/2016 virus, in group B, five goats were infected with BTV-25-GER2018 virus and in group C, five goats were infected with BTV-8-GER2018 virus. Back titration on BSR cells revealed the following titres:
For group A, BTV-33-MNG3/2016, 10^4.83^ CCID_50_/mL (per goat 2.7*10^5^ CCID_50_);For group B, BTV-25-GER2018, 10^3.3^ CCID_50_/mL (per goat 8.0 * 10^3^ CCID_50_);For group C, BTV-8-GER2018, 10^3.67^ CCID_50_/mL (per goat 1.9*10^4^ CCID_50_). 

Body temperature was taken daily, and the goats were clinically scored throughout the experiment, applying a modification of the Clinical Score system [31] described in an earlier study [7]. On the sampling days (0, 3, 5, 7, 10, 12, 14, 17, 21, 24, 28, 31, 35 and 42 dpi), nasal, oral, ocular and rectal swabs were taken as well as blood from the jugular vein by using an adapter system (Kabe Labortechnik GmbH, Nümbrecht, Germany). We used plane tubes containing ethylenediaminetetraacetic acid (EDTA) for EDTA blood collection and serum tubes without anticoagulants for serum collection (Kabe Labortechnik GmbH, Nümbrecht, Germany). Serum was collected after centrifugation at 2000 rpm for 15 min. After euthanising, lung, liver, spleen, mediastinal and mesenterial lymph nodes were taken from the goats positive for the BTV genome in the EDTA blood by RT-qPCR. (Once a goat was noted as positive in the study, it remained classified as positive until the end of the study.)

### 2.4. RNA Extraction and RT-qPCR

Lentil-sized organ samples (approx. 30 mg) were homogenised in 500 µL of a serum-free medium using the TissueLyser II tissue homogeniser (QIAGEN, Hilden, Germany). For RNA extraction, 100 µL of the liquid starting material was used (cell culture material, EDTA blood, serum, swab samples collected in 2 mL serum-free media and the liquid part of the homogenised organ material) and eluted in 100 µL of elution buffer. Extraction was performed with the NucleoMagVET kit (Macherey-Nagel, Düren, Germany) using the half-automated KingFisher platform (King-Fisher Flex magnetic particle processor, Thermo Fisher Scientific, Darmstadt, Germany). As a control for successful RNA extraction, 10 µL of internal control RNA (IC-2 RNA) was added during the extraction process [32]. The RNA was amplified with the Pan-BTV-S10-RT-qPCR recommended by the OIE [33], adapted with an additional probe and named BTV-S10-primer-probe-mix-v2. Comparative evaluation data for the BTV-S10-OIE assay and the modified BTV-S10-primer-probe-mix-v2 are summarised in the Appendix A. For the generation of 200 µL of BTV-S10-primer-probe-mix-v2, the following oligos were mixed: 20.0 µL of BTV_IVI_F (5’-TGG AYA AAG CRA TGT CAA A-3´), 20.0 µL of BTV_IVI_R (5´-ACR TCA TCA CGA AAC GCT TC-3´), 3.75 µL of BTV_IVI_FAM (5´-FAM-ARG CTG CAT TCG CAT CGT ACG C-BHQ1-3´), 2.50 µL of BTV_IVI_FAM_v2 (5´-FAM-AGG CTG CAT ACG CAT CRT ACG C-BHQ1-3´) and 153.75µL of 0.1x TE (pH 8.0). The final composition of the RT-qPCR reactions was 1.25 μL of RNase-free water, 6.25 μL of 2x RT-PCR buffer, 0.5 μL of RT-PCR Enzyme Mix, 1 μL of BTV-S10-primer-probe-mix-v2-FAM, 1 μL of EGFP-mix1-HEX and 2.5 µL of the heat-denatured template RNA (at 95 °C for 5 min). All RT-qPCRs were run on the CFX 96 real-time PCR cycler (Bio-Rad, Hercules, CA, USA) with the AgPath-ID™ One-Step RT-PCR Reagents from Applied Biosystems™ (Waltham, MA, USA). The temperature profile used was 10 min at 45 °C (reverse transcription) and 10 min at 95 °C (inactivation of the reverse transcriptase/activation Taq polymerase) followed by 42 cycles of 15 s at 95 °C (denaturation), 20 s at 56 °C (annealing) and 30 s at 72 °C (elongation). Fluorescence values (FAM, HEX) were collected during the annealing step. Samples were considered positive when the quantification cycle (Cq) values were <40.

In every run, a generated BTV standard series produced by droplet PCR (QX200 Droplet Digital PCR System, Bio Rad, Hercules, CA, USA) was included for calculating the genome copy numbers.

### 2.5. Serological Analysis

#### 2.5.1. ELISA

Serum samples of 0, 7, 14, 21, 28 and 42 dpi were screened for BTV-group-specific antibodies (VP7) using a competitiveELISA (ID Screen^®^ Bluetongue Competition, ID-Vet, France) and plate reader reading at a wavelength of 450 nm according to the manufacturer’s instructions. The results were expressed as the percent of negativity compared to the negative kit control (% S/N = optical density (OD) of the sample/OD of the negative control multiplied by 100) and denoted as a positive or negative result (<50% S/N were considered as positive, samples with ≥50% S/N as negative).

#### 2.5.2. VNT

Serum samples of 0, 7, 14, 21, 28 and 42 dpi of goats positive for BTV RNA in the EDTA blood were screened by VNT for the presence of neutralising antibodies.

Briefly, the serum was diluted in log2 steps starting from 1:10 to 1:1280 and titrated against 100 CCID_50_ of the respective virus (BTV-25-GER2018, BTV-33-MNG3/2016 or BTV-8-GER2018). The plates were incubated for 1 h at 37 °C before overnight incubation at 4 °C. The following day, 100 µL of a BHK-21 (BSR/5) cell suspension of approximately 30,000 cells/100 µL was added per well. After incubation for 3–5 days at 37 °C, the plates were screened with a stereomicroscope for cytopathic effect (CpE). The neutralisation titre was determined as the highest dilution of serum with 100% neutralisation (no CpE). The Spearman and Kärber methods were used for the calculations.

### 2.6. Isolation in Cell Culture

All the blood samples from the experimentally infected goats were processed identically for the virus-isolation experiments: 500 µL of EDTA blood was centrifuged (8000 rpm) for 2 min, and the red blood cells were washed twice in 1 mL of PBS and finally diluted in 500 µL of PBS. In cases of no sufficient cell pellet being generated after initial centrifugation, the unwashed blood was used for virus isolation experiments. In both cases, blood preparations were lysed using 20 s ultrasound treatment at 30 W (Sonifier 450, Branson Ultrasonics, Danbury, CT, USA). BHK-21 (BSR/5) cells in T25 cm² cell flasks were initially incubated for three hours at 37 °C using the cultivation medium with 5.32 g of Hanks salts, 4.76 g of Earle’s salts, 1.25 g of NaHCO3, 10 mL of non-essential amino acids and 120 mg of sodium pyruvate per litre of medium (FLI intern medium number ZB5d) supplemented with 10% foetal calf serum (FCS). Afterwards, the cells were inoculated with 500 µL of the blood preparation for two hours and incubated on a tilt shaker in an incubator at 37 °C and 5% CO_2_. Then, the blood inoculum was removed, and the flasks were refilled with the medium supplemented with 10% FCS and antibiotics at double the standard concentration (20,000 µg/mL Penicillin, 20,000 units/mL Streptomycin, 10 mg/mL Gentamicin, 250 µg/mL Amphotericin B). After 3 to 4 days of incubation at 37 °C, the infected BSR cell monolayer was split by using 1 mL of trypsin and mixed with 5 mL of the supernatant. In the next step, 3 mL of the cell–trypsin–supernatant suspension was transferred to a new T75 cm² cell flask with fresh BSR cells grown for 3 h. Three passages were performed, and the success of virus replication was confirmed by the genomic load estimated by RT-qPCR.

### 2.7. Blood Analysis

For the blood analysis, the EDTA blood was analysed within the same day of sampling using the ProCyte Dx (Idexx, Westbrook, ME, USA) according to the manufacturer’s instructions. Several blood parameters were measured including the total white blood cell count, lymphocytes, neutrophils, monocytes, basophils, thrombocytes, erythrocytes and haematocrit.

### 2.8. Statistical Analysis

For statistical analysis, we used the Mann–Whitney U test with Bonferroni correction for the individual bleeding time points in GraphPad Prism 9.0.0. We compared the viraemic animals of groups A and B infected with the atypical BTV strains (*n* = 6) and the viraemic animals of group C infected with the classical BTV strain (*n* = 2).

Viraemia was analysed by comparing the dpi of its onset, the maximum viral load in log_10_ genome copies/mL and the total virus production with the area under the curve (AUC). Furthermore, we analysed the total WBC count, the lymphocyte count, the monocyte count, basophils, thrombocytes and erythrocytes in K/µL (1000 cells per µL) and haematocrit in %. Non-parametric tests were preferred because of the small group sizes and the potential non-normality of the data.

## 3. Results

### 3.1. Clinical Manifestation

The animals remained healthy throughout the whole study period. No fever or rise in temperature was recorded (body temperature < 40 °C).

### 3.2. RT-qPCR Results

#### 3.2.1. EDTA Blood

All RT-qPCR results are shown in Figure 1. All in-contact goats of all three groups (A, B and C) remained negative in RT-qPCR throughout the complete animal trial.

In group A, all five goats inoculated with BTV-33-MNG3/2016 yielded positive results in the PAN-BTV-S10_v2, starting from 5 to 10 dpi until the end of the trial at 42 dpi. BTV genome detection in goats A/32 and A/54 started at 5 dpi with Cq values of 35.0 and 35.6 (1.0 and 0.8 log_10_ genome copies/mL). The peak Cq value of goat A/32 was reached at 14 dpi with 27.0 (3.7 log_10_ genome copies/mL), and that of goat A/54 was reached at 10 dpi with 27.2 (3.5 log_10_ genome copies/mL). For the goats A/36 and A/24, the BTV genome was first detected at 7 dpi with Cq values of 35.7 and 36.0 (0.8 and 0.7 log_10_ genome copies/mL) and peak Cq values at 12 dpi, reaching 27, and at 14 dpi with 22.5 (3.6 and 5.1 log_10_ genome copies/mL). Goat A/23 was detected to be positive for the BTV genome at 10 dpi with a Cq value of 36.6 (0.5 log_10_ genome copies/mL) and peaked at 17 dpi with a Cq value of 25.9 (4.0 log_10_ genome copies/mL). At the end of the trial at 42 dpi, the PAN-BTV-S10_v2 results of the five infected goats ranged between 28.2 and 31.9 (2.2–3.4 log_10_ genome copies/mL). 

In group B, one of the five goats inoculated with BTV-25-GER2018 was detected to be positive for the BTV genome during the animal trial. The other four inoculated goats remained negative in the PAN-BTV-S10_v2 RT-qPCR. The positive goat B/021 started with a Cq value of 31.5 at 12 dpi (2.4 log_10_ genome copies/mL), peaked at 17 dpi with a Cq value of 27.5 (3.5 log_10_ genome copies/mL) and ended with 32.9 at 42 dpi (1.9 log_10_ genome copies/mL).

In group C, two of the five goats inoculated with BTV-8-GER2018 were detected to be positive for the BTV genome starting at 10 dpi until 42 dpi, whereas the other three inoculated goats stayed negative in the RT-qPCR. Goats C/28 and C/43 started at 10 dpi with Cq values of 37.2 and 33.2 (0.3 and 1.6 log_10_ genome copies/mL). Goat C/28 peaked with a Cq value of 28.2 at 17 dpi, and goat C/43 peaked with a Cq value of 27.3 at 14 dpi (3.4 and 3.6 log_10_ genome copies/mL). At 42 dpi, the Cq value of goat C/28 decreased to 34.8, and that of goat C/43 decreased to 36.9 (1.3 and 0.7 log_10_ genome copies/mL).

Comparing the onset of viraemia in the dpi of the goats infected with the atypical BTV strains (median of 7) to that of the goats infected with the classical BTV strains (median of 10), we did not find a significant difference (*p* = 0.39); the same applies to the maximum viral load in log_10_ genome copies/mL in atypical BTV strains (median 3.65) and classical BTV strains (median 3.5) (*p* = 0.36). The total virus production analysed by using the AUC of the goats infected with the atypical strains (median of 18,449) did not differ significantly (*p* = 0.14) from that of the goats infected with the classical BTV strains (median of 7275).

#### 3.2.2. Serum

Only 34 serum samples were positive in the RT-qPCR (in comparison to 85 positive EDTA blood samples), with less genome copies/mL compared to the EDTA blood results. The ΔCq values of the respective EDTA blood samples and the serum samples varied from 5.4 to 11.5 (2.5–4.8 log_10_ genome copies/mL; data not shown).

#### 3.2.3. Swabs

All the nasal, oral, ocular and rectal swabs of all the sampling days during the animal trial were consistently negative for the BTV genome in RT-qPCR.

#### 3.2.4. Organs

During necropsy, none of the analysed organs showed macroscopical lesions. The RT-qPCR results of the organ samples of the infected animals are shown in Table 1. The spleens were all positive in the PAN-S10 RT-qPCR and showed the lowest Cq values in comparison to the other organs of each individual goat. For groups A and B, the Cq values varied between 28.4 and 31.2 (3.1 and 2.4 log_10_ genome copies/mL), and in group C, Cq values of 32.3 and 34.5 were reached (2.1 and 1.5 log_10_ genome copies/mL). For groups A and B, the liver, lung and mediastinal lymph node samples were positive for the BTV genome, whereas these organ samples were negative for the BTV genome in the tested BTV-8-infeCqed goats C/28 and C/43 in group C. The mesenterial lymph node samples were negative for the BTV genome in all tested animals.

### 3.3. Serological Analysis

#### 3.3.1. ELISA

All the cELISA results are shown in Figure 1. In groups A and B, all the goats remained negative in the cELISA during the entire animal trial. However, in group A at 42 dpi, the cELISA results slightly approached the cut-off line. The goats in group C tested positive for BTV RNA in the EDTA blood, started to be highly positive at 21 dpi in the cELISA and remained positive until the end of the animal trial.

#### 3.3.2. VNT

The tested sera of goats infected with BTV-33-MNG3/2016 and BTV-25-GER2018 did not neutralise the respective virus in the VNT from 0 dpi until the end of the animal trial at 42 dpi. However, the sera of goats C/28 and C/43 neutralised BTV-8-GER2018, starting from 21 dpi until 42 dpi. In detail, for goat C/28 at 21 dpi, the 50% neutralisation dose (ND_50_) was 28, increasing to 36 at 28 dpi and to 90 at 42 dpi. For goat C/43 at 21 dpi, the ND_50_ was 14, rising to 180 at 28 dpi and to 143 at 42 dpi.

### 3.4. Virus Isolation

BTV-33-MNG3/2016 was reisolated from the EDTA blood of all five RT-qPCR positive goats, starting from 7–12 dpi until 17–31 dpi (mean isolation period of 13.6 days; standard deviation (SD) of 5.2). For BTV-25, we reisolated the virus at 17 dpi only. BTV-8 was reisolated from the two RT-qPCR-positive goats starting from 12–14 dpi until 17–21 dpi (mean of 7 days; SD 1.4).

### 3.5. Blood Analysis

The results of the total WBC, lymphocyte and monocyte counts measured during the animal trial are shown in Figure 2. Here, a rise in WBC, lymphocytes and monocytes for all the goats infected with the atypical BTV (Group A and B) can be seen, whereas for the BTV-8-infected goats in group C, no increase was measured over the whole study period.

Comparing the total WBC count of goats infected with the atypical BTV strains (median of 19.7) to the classical BTV-8 strain (median 9.44) during all 15 blood sampling times (α = 0.0033), the WBC count was significantly higher for the atypical BTV strains (*p* < 0.0001). The differences between the total WBC count of the 14 blood sampling time points compared to the 0 dpi of the respective animals (α = 0.0036) were significantly higher for the atypical BTV strains (*p* = 0.0035) compared to the classical BTV group.

Comparing the total lymphocyte count of goats infected with the atypical BTV strains (median of 10.14) to the classical BTV strains (median 6.84) during all 15 blood sampling times (α = 0.0033), the lymphocyte count was significantly higher for the atypical BTV strains (*p* < 0.0001). However, no significant differences (α = 0.0036) between the atypical BTV and the classical BTV group were seen when the 14 lymphocyte blood sampling values were compared to the 0 dpi value of the respective animals (*p* = 0.29).

For the total monocyte count of goats infected with the atypical BTV strains (median of 4.08) to the classical BTV strains (median 0.83) during all 15 blood sampling times (α = 0.0033), the monocyte count was significantly higher for the atypical BTV strains (*p* < 0.0001). No significant differences (α = 0.0036) were observed when the 14 monocyte blood sampling values were compared to the 0 dpi value of the respective animals (*p* = 0.026).

No significant differences were seen for neutrophils, basophils, thrombocytes, erythrocytes and haematocrit between the atypical and classical groups.

## 4. Discussion

Atypical BTV has been on the rise in the last few years since the first discovery of BTV-25-TOV in 2008. Investigations on the pathogenesis of different BTV strains were conducted over the years, for classical as well as atypical BTV strains. In our study, we investigated the comparative pathogenesis and immune response of classical and atypical BTV strains using three different goat groups. To represent the classical BTV, we chose the re-emerged BTV-8 strain isolated from cattle in Germany, whose kinetics and pathogenesis in goats have never been tested before. To represent atypical BTV strains, we chose BTV-25 isolated from goats in Germany and BTV-33 isolated from goats in Mongolia. All three strains were isolated from asymptomatic ruminants naturally infected in the field. The pathogenesis and immunological study revealed clear differences concerning the measured blood parameters such as total WBC, lymphocyte and monocyte count. Furthermore, only the animals infected (two of the five inoculated goats) with the classical BTV strains showed a humoral response. However, the atypical and classical strains did not diverge regarding the dpi at which viraemia began, the maximum viral load in log_10_ genome copies/mL and the total virus production.

The clinical picture of BTV depends not only on the virus strain but also on the affected host species [10]. Severe disease is reported for sheep, whereas for other domestic ruminants, such as goats and cattle, subclinical manifestation is stated [15]. The atypical virus strains chosen for the animal trial were isolated from asymptomatic goats. This is in line with the findings of the animal trial performed here, where all goats infected with BTV-25-GER2018 and BTV-33-MNG3/2016 showed an asymptomatic BTV infection. For BTV-8, however, clinical disease was seen in cattle and goats during the first outbreak from 2006 to 2009 and with less severity in cattle in the second BTV-8 epidemic in 2015–2019 [11,34]. During the first epidemic, goats showed a high seroprevalence rate in northwestern Europe (25% in Germany) [19], and mild clinical disease could be seen in one experimental BTV-8-infected goat with fever and generalised illness including apathy, dysphagia, diarrhoea and lameness [35]. Furthermore, BTV-8 was able to cross the caprine placenta, as reported for cattle [36]. To our knowledge, we have tested the newly emerged BTV-8 strain (BTV-8-GER2018) for the first time in goat as the host species. The two infected goats in our study showed neither clinical signs nor fever, which is line with reports about subclinical disease in goats and a reduced pathogenicity of the newly emerged BTV-8 strain [23].

In our animal experiment, we observed a difference between the inoculated and infected goats. For BTV-33-MNG3/2016, five out of five inoculated goats became infected, whereas for BTV-8-GER2018, two became infected, and for BTV-25-GER2018, only one goat became infected out of five. An experimental study with BTV-8 in cattle did not find a correlation between the amount of inoculum and the kinetics of viraemia [37]. In this study, the length and the intensity of viraemia and the neutralising antibody response did not differ between the different infectious doses, varying from just 10 TCID_50_ to 10^6^ TCID_50_ [37]. Through the natural inoculation route by BTV-positive *Culicoides* midges, only one bite can transmit around 0.32 to 7.79 TCID_50_ and establish infection [38,39]. The probability of virus transmission from an infectious vector to a susceptible ruminant host was described as close to 100% [40]. The inoculation titres used in our study are all in the range of other animal trials, and lower infection titres more likely resemble the natural vector-borne transmission cycle. However, we did not use the same infectious dose for all groups, and it cannot be completely ruled out that it had an influence on the study outcome. The reduced infectivity of BTV-8-GER2018 after subcutaneous infection described in our study (two of five goats became BTV-positive) might be in line with the findings of Flannery et al. 2019, who described the re-emerged BTV-8 strain as having a reduced vector competence as well as the relatively slow spread of the re-emerged BTV strain [23]. For BTV-25-GER2018, a moderate seroprevalence of 18% to 23% was described in a naturally infected goat flock over a prolonged period of time (4.5 years), whereas 31% to 38% were BTV RNA-positive [27]. These results of only moderate seroprevalence over a longer time period, together with the findings of our study that only one of five goats became BTV-positive after inoculation, suggest a low infectivity of BTV-25-GER2018. For BTV-33-MNG3/2016, no data are available about the seroprevalence or infectivity of this atypical BTV strain in the field. Hence, our study indicates a potentially high infectivity rate of BTV-33-MNG3/2016 in goats.

Interestingly, BTV-8-GER2018 and BTV-25-GER2018 showed moderate Cq values with peaks between 27.3 and 28.2 (3.4–3.6 log_10_ genome copies/mL) in the EDTA blood. These relatively high Cq value peaks are not surprising for atypical BTV strains but are surprising for BTV-8. However, our results of a reduced viraemia of the German re-emerged BTV-8 strain are in line with the observed lower genome concentrations seen in BTV-8-FRA2017-infected sheep in comparison to the UKG2007 strain. Remarkably, BTV-33-MNG3/2016 showed the lowest peak Cq values (two of the five goats, 22.5 and 25.9). Nevertheless, BTV-33-MNG3/2016 was not the only atypical BTV strain that showed lower Cq values during infection, comparable to classical BTV strains; this has also been described for BTV-26, BTV-27 and BTV-28 [25,28,29]. The high ΔCq value differences among the respective serum samples throughout the study period indicate that BTV is connected to the red blood cells, and only few virus particles circulate freely. For the classical BTV strains, this phenomenon is well known [41], and our study confirms it for the two atypical BTV strains tested here as well. The time point of infection varied little in the groups, starting with the earliest from 5 to 10 dpi with BTV-33-MNG3/2016 followed by BTV-8 at 10 dpi and BTV-25 at 12 dpi. For the BTV-8 strains FRA2017 and UKG2007, earlier infection time points starting at 2 dpi have been described in sheep [23]. Hence, our findings for BTV-8-GER2018 presenting a late onset of viraemia are surprising and prove that BTV can vary strongly within the same serotype. However, no significant difference regarding the onset of viraemia between the atypical and classical strains was found in our study. For all three groups, no contact transmission was observed, and, therefore, this does not seem to play a role for BTV-25-GER2018, BTV-33-MNG3/2016 and BTV-8-GER2018. We were able to reisolate all three virus strains from the RT-qPCR animals. For successful virus isolation, the Cq value seems to play a crucial role. This could explain the shorter time periods of virus isolation for BTV-8 (7 days; SD 1.4) and BTV-25 (1 day), with higher Cq values, compared to BTV-33-MNG3/2016 (13.6 days; SD 5.2), with lower Cq values. The virus isolation length of BTV-8-GER2018 is consistent with the findings of related BTV-8 strains [23].

For all three virus strains, the spleens at 42 dpi showed the lowest Cq values. Lung, liver and mediastinal lymph nodes were positive for BTV RNA for the two atypical BTV strains but not for BTV-8. One hypothesis might be the ascending humoral immune response of BTV-8, starting to clear BTV infection in the case of BTV-8. In detail, the two goats infected with BTV-8, which were positive in the RT-qPCR, developed BTV group- and serotype-specific antibodies starting at 21 dpi, whereas the goats infected with BTV-25 and BTV-33-MNG3/2016 did not develop group- and serotype-specific antibodies, despite positivity in the RT-qPCR. However, at 42 dpi, there was a very slight tendency of goats positive for BTV-25 and BTV-33-MNG3/2016 in the RT-qPCR to approach the cut-off value in the cELISA. A longer study period than 42 dpi might have shown whether these goats infected with atypical BTV would have become cELISA-positive. However, the humoral immune response of goats infected with the atypical BTV strains was not comparable to the fast and strong reaction of goats infected with BTV-8 in our study. We can conclude that there is a difference in the immunopathogenesis of the classical BTV strains and the atypical BTV strains BTV-25-GER2018 and BTV-33-MNG3/2016 tested here.

The difference in immunopathogenesis is underlined by the blood parameters measured during this study. For goats infected with BTV-33-MNG3/2016 and BTV-25-GER2018, we stated a significant difference in the WBC count compared to the goats infected with the classical strain. In general, the WBC count of goats is described as 9000 cells/µL with a range of 4000 to 14,000. Leukocytosis starts at a WBC count of more than 13,000/µL, and leukopenia starts at less than 4000 /µL. However, total WBC counts and differential cell counts vary significantly with age: for 3-month-old goats, increased WBC counts of 18.18 ± 3.84 × 10^3^ /µL were reported, whereas for adult goats at an age of 2 years, the WBC count decreased to 8.08 ± 2.51 × 10^3^/µL [42]. Hence, our 6-month-old goats in the animal trial showed higher WBC counts than adults, but the values are comparable to those of the 3-month-old goats described in the literature. Nevertheless, the goats infected with atypical BTV strains developed a significant increase in WBCs over the course of infection, which can be constituted as leukocytosis from approx. 17 dpi until the end of the animal trial at 42 dpi. In detail, we observed lymphocytosis and monocytosis in all of the viraemic animals infected with the atypical strains. BTV infection was previously described to be followed by pan-leukopenia, with its peak at 7–8 dpi [5]. Transient leukopenia and lymphopenia were seen in previous studies with BTV-1 and BTV-8 in sheep, where leukopenia occurred during the very first days of infection, resulting from a depletion of T-lymphocytes. However, in the same study, a proliferation in the B cell population was seen, possibly linked to the upcoming antibody production [43]. However, in our study, we found leukopenia neither during the first days after inoculation nor over the whole time period. It remains unknown which lymphocyte fraction increased in our animal study, and it is difficult to link the increase to the B cell fraction, as the antibody levels did not rise in groups A and B. An increase in monocytes has not been commonly described in other BTV infection studies [43]. In future studies, more details about the alterations in the lymphocyte populations, as well as various cytokines, should be collected to understand the differences in immunopathogenesis between classical and atypical BTV serotypes.

In conclusion, our animal study showed differences in the immunological reaction between the two atypical BTV strains BTV-25-GER2018 and BTV-33-MNG3/2016 and the classical BTV strain BTV-8-GER2018. We found differences in the humoral response and also in the total WBC count, which indicate a possible difference in the cellular immune response. However, regarding the infectivity rate and peak Cq values, the atypical and classical BTV strains did not differ. Our experiment underlines that BTV has a highly variable disease outcome depending on the strains, the host and the individual differences between animals, which makes the generalisation of our results (atypical versus classical BTV strains) difficult. Nevertheless, our animal study strongly suggests that the phylogenetically distinct atypical BTV strains differ from the classical BTV strains in terms of immunopathogenesis. Furthermore, no horizontal transmission was seen for BTV-25, BTV-33 and BTV-8.

## Figures and Tables

**Figure 1 viruses-14-01034-f001:**
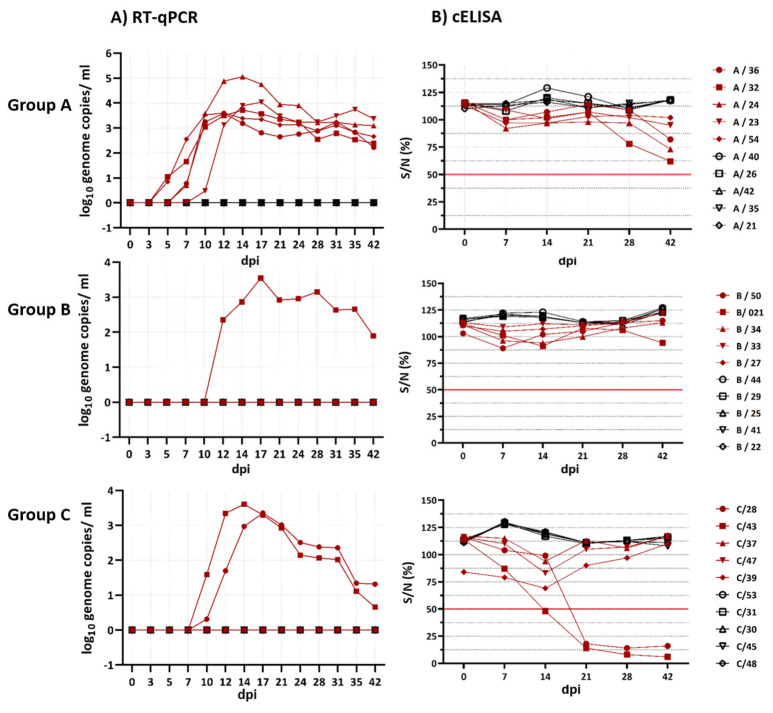
In total, five goats of group A were inoculated subcutaneously with BTV-33-MNG3/2016, five goats of group B with BTV-25-GER2018 and five goats of group C with BTV-8-GER2018 (in red). In each of the three groups, five control goats were kept (in black). (**A**) RT-qPCR results of the EDTA blood. (**B**) The cELISA results of the serum. The horizontal red line is the cut-off line (values ≤ 50 are considered positive; values ≥ 50 are considered negative). dpi = days post infection; S/N (%) = signal-to-noise ratio in %.

**Figure 2 viruses-14-01034-f002:**
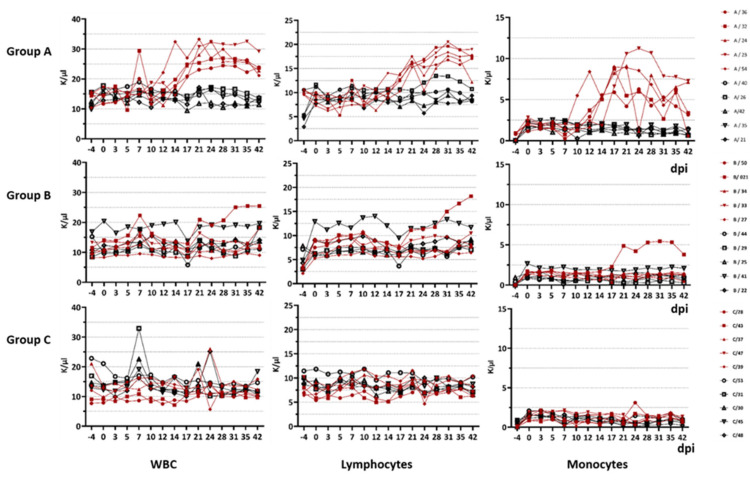
Total WBC count, total lymphocyte count and total monocyte count of the EDTA blood.

**Table 1 viruses-14-01034-t001:** RT-qPCR results (Cq values = cycle of quantification) of the organs of the viraemic goats during experimental infection. Organs were taken during necropsy at 42 dpi.

Inoculated Virus	Goat ID	Lung	Liver	Spleen	Mediastinal Lymph Node	Mesenterial Lymph Node
BTV-33-MNG3/2016	A/36	34.59	35.15	29.98	36.64	no Cq
	A/32	31.4	33.99	30.4	36.87	no Cq
	A/24	31.13	33.86	31.24	37.68	no Cq
	A/23	30.23	32.6	28.38	36.54	no Cq
	A/54	38.94	35.24	31.06	37.55	no Cq
BTV-25-GER2018	B/021	34.43	31.64	30.78	38.17	no Cq
BTV-8-GER2018	C/28	no Cq	no Cq	34.49	no Cq	no Cq
	C/43	no Cq	no Cq	32.27	no Cq	no Cq

## Data Availability

The data presented in this study are available in the article and Appendix A here.

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
