# Peer review of "Bluetongue Virus Infection of Goats: Re-Emerged European Serotype 8 vs. Two Atypical Serotypes"

_viruses, 2022, doi:10.3390/v14051034_

Round 1

Reviewer 1 Report

General comments:

The manuscript would greatly benefit from an English language editor.

Please ensure that all abbreviations have been described at first use.

Immune response rather than pathogenesis is the focus of this article and the title should be changed accordingly.

Specific comments:

Line 32 – BTV not described at first use.

Line 36 – The use of ‘serotyped’ serotypes is confusing. In the previous sentence, neutralization by reference serum is said to classify the notifiable 24 serotypes. But here 27 serotypes are said to be identifiable by this method. Are the 3 extra serotypes, identified by neutralization or are they just not notifiable?

Line 37 – Arbovirus is not described at first use. This sentence is awkwardly constructed and the fact that infected female Culicoides are the biological vector of bluetongue virus is lost.

Line 40 – Does bluetongue here refer to the virus or the disease?

Line 52-53 – The information about cattle and goats typically showing subclinical disease would be better placed in the paragraph above.

Line 50, 60 – Germany is said to be part of northern Europe in the first line and central Europe in the second.

Line 67-68 – Does this mean atypical BTV has not been identified in cattle or had not been identified in cattle before a certain time?

Line 69 – How is viremia detected over these long periods of time? Is it by viral isolation or by RT-qPCR? One suggests active replication, the other does not.

Line 72 – VNT not described at first use. This sentence is out of place here.

Line 74-75 – This sentence is not informative and should be deleted.

Line 85 – Horizontal transmission can be in several ways. The extra goats were used to examine direct or contact transmission.

Materials and Methods

Line 100 –What was the target for the cELISA? Previously it was said that atypical BTVs do not produce robust neutralizing antibodies.

Line 106 – How are BHK-21 [CT] cells different from BHK-21 [BSR/5] cells?

Line 116 – Where were the goats injected?

Line 124 – EDTA not spelled out on first use.

Line 125 – Serum cannot be directly taken from an animal. How were serum samples obtained?

Line 126 – At what point were the goats positive by RT-qPCR? The sampling day prior to euthanasia? At least once in the study?

Line 136 – What was the internal control RNA that was used? How much was added to the extraction? What Ct was considered a successful RNA extraction?

Line 138 – Please add the details of how the addition of the extra primers and probe affected the reaction efficiency and sensitivity, limit of detection. This can be added as a supplementary file.

Line 139 – Please make a table of the primers and probes used in the reaction including those for the internal control. Please add the final concentration of each in the reaction.

Line 145 – Please make sure that the primer and probe mix is labeled the same throughout the paper.

Line 146 – Was template RNA heat denatured prior to addition? For how long and at what temperature? How many technical replicates were run for each sample?

Line 152 – Plates are usually read during the extension step. Why was 40 chosen as a cutoff?

Line 156 – What is the BTV-group-specific target? VP7? Please briefly describe the assay. Was this a colorimetric assay or fluorescent? Was it read by eye or on a plate reader at a specific wavelength? Was the cELISA for IgG or IgM antibodies?

Line 158-159 – This is very unclear.

Line 162 – VNT not described at first use.

Line 166 – Which cells are these? BHK-21?

Line 167 – Were wells fixed and stained before scoring or was CpE observed under a microscope.

Line 168 – 100% neutralization or no CpE.

Line 175 – Please identify which samples produced no cell pellets.

Line 179 – What was the concentration of the essential amino acids?

Line 181 – Was the 2 hour incubation in an incubator? At what temperature, CO2 %? Were the plates rocked?

Line 185 – This sentence is unclear. The monolayer was dissociated with trypsin? Why would the cells be mixed with supernatant rather than fresh medium?

Line 204 – K/ul not described at first use.

Results

There is no standard curve described in the methods section or mention of including standards in the RT-qPCR plates. Please explain how the genome copies/ml were calculated.

It is more informative to present the genome copies/ml values in this section than the CT values. Presenting both makes it difficult to keep the numbers straight.

Line 241 – Please add supplementary table of the area under the curve values for each group.

Figure 1: Separate into two figures or label A) RT-qPCR, B) cELISA

Please add a statement in the caption for the cELISA data that above the line is negative and below the line is positive. Without knowing the specifics of this test it could be easily misinterpreted.

Please make the middle graph of the cELISA data have the same X-axis as the others.

Since the uninoculated animals did not show any evidence of infection, it would make the graphs easier to read if they were removed.

There is no mention of this figure for the cELISA data.

Line 251 – A supplementary table showing the genome copies/ml values for blood and serum for each animal would be useful to other researchers.

The delta Ct values presented are hard to interpret. How long was virus detectable in the serum versus in blood? Was virus detectable in serum before or after it was detectable in the blood?

Line 261 – How can you calculate genome copies/ml from organ samples? Shouldn’t they be in genome copies/mg or g? The only indication of size was that they were lentil size.

Line 278-283 – ND50 not described at first use. Please use either the ND50 or the dilution value.

Line 286 – Please use 7 to 12 or 7 – 12 but not both.

Line 286 – What is meant by “Animals were positive”? Virus was able to be isolated from day x to day y?

Figure 2: The uninoculated animals in these graphs make it very difficult to differentiate the different animals that actually had a response.

It’s understandable why the two atypical BTV strains are grouped together for analysis but it may be informative to compare the inoculated animals and the uninoculated animals in each group.

Discussion

Line 318 – TOV not described at first use.

Line 320 – The study is more focused on immune response than pathogenesis.

Line 328 – Two of the five inoculated showed an antibody response

Please make sure that it is clear that your statements refer to those goats that showed evidence of infection rather than all inoculated goats. The reader may unintentionally assume inoculated = infected.

Line 360 – Infectivity does not always equal high pathogenesis. It may be that subcutaneous injection results in reduced infectivity.

Line 365 – How long is a prolonged period of time? Months? Years?

Line 372 – These Ct values are for what samples? Were genome copy numbers available for these samples?

Line 382 – Is this true throughout the study period? or at specific time points?

Reviewer 2 Report

The paper provides additional information about the kinetics and pathogenesis of a remerging BTV8 and two atypical BTV strains (25 and 33) and highlights the differences among them. Moreover, as far as I know is the only recent paper describing the effect of BTV on white blood cells count. I also appreciate the idea to compare the Cq results between EDTA blood and serum, showing that also atypical BTVs are “stuck” to RBCs.

Discussion section cover all aspects of the results.

Overall the manuscript would benefit from some editorial minor editorial language.

Title

-I suggest to find a more engaging title, but this is an issue Authors should discuss with the editorial office.

-Delete “of” after Bluetongue virus

Abstract

Line 14: ...the study to detect a potential contact transmission of the virus

Line 15: After infection, Edta...

Line 15: were ocular swabs also collected as reported in M&M?

Line 16: …analyses, respectively

Line 16: delete “animal”

Line 21: …atypical BTVs used in the trial differ…

Line 22: …, no evidence of direct contact transmission was found.   

Introduction

In my opinion this section is too long and would benefit of a cut, for example deleting/shortening the part on the clinical symptoms and on the incursion of BTV-8 2006-2009.  

Line 87: as no consensus is reached on the numbering of BTV serotypes, please refer to BTV-33 as putative BTV-33, just at first appearence in the text.

M&M

2.1

Lines 94-95: …3 conditions (twenty-six female…male goats)

Line 99-100: were the goats tested by PCR? Please specify

2.2

Lines 102-103: …healthy goats, as well as…, were chosen for…

2.3

Line 119: typo - “with”, not “wit”

Back-titration: the inoculated dose is much higher for BTV33 than for BTV25 and 8 (34- and 14-folds, respectively). This is a huge difference when it comes to clinical outcome and infection kinetics. Did the authors titrated the virus before inoculation? Why the authors did not “normalize” the injected infectious dose (for example by diluting btv33 1 to 10 and then administering different volumes)?

Line 121: temperature was taken daily throughout the trial? I cannot find any reference to body temperature in the Results section. Was temperature always in the normal range for the species? Which was the cutoff fever/not fever?

Lines 125-126: delete “several organ samples”

Line 126: …taken from the goats

2.5.1

Why the authors didn’t test by ELISA all the samples (all days)

2.5.2

Why the authors didn’t test by VNT all the samples (all days)

Line 162: …were screened by…

Line 168: …as the highest dilution…

Discussion

Line321: …classical BTV we chose (or selected) the remerged…

Lines 322-323: ...cattle in Germany, whose kinetics and pathogenesis in goats was never tested before

Overall, I suggest to check the Discussion section for typos and minor grammar errors

Round 2

Reviewer 1 Report

The authors' have answered my comments and concern to my satisfaction. 

The English has improved a great deal in this version.

Reviewer 2 Report

I read carefully the Authors' response to my and other reviewer's comments. I cheched again the revised manuscript. All my comments have been addressed carefully. I have no more concern about the manuscript and I am pleased to accept the manuscript in its present form. 

I do not agree with some of the other reviewer's comments. However, the Authors respectfully did not fulfill all his requests.   

I again underline my appreciation for the idea to compare the Cq results between EDTA blood and serum, showing that also atypical BTVs (or at least the one studied in the paper) are “stuck” to RBCs. This is a useful information for BTV community and a starting idea for other studies.

Discussion section cover all aspects of the results.

Just a remark: Line162: colorimetric or competitive ELISA?